# Predictors associated with Clavien–Dindo complications in lung cancer surgery: A retrospective cohort study

Mantana Saetang[1], Thitikan Kunapaisal[1]\*, Wirat Wasinwong[1], Parin Boonthum[2], Bussarin Sriyanaluk[1], Kanjana Nuanjun[1]

**1** Department of Anesthesiology, Faculty of Medicine, Prince of Songkla University, Hat-Yai, Songkhla, Thailand, **2** Division of CardioVascular Thoracic Surgery, Department of Surgery, Faculty of Medicine, Prince of Songkla University, Hat-Yai, Songkhla, Thailand

\* thitikan070@gmail.com

## Abstract

### Background

To highlight the risk assessment tool associated with postoperative cardiopulmonary complications of Clavien–Dindo (CD) $\geq$ II in elderly patients who underwent lung cancer surgery.

### Methods

In patients $\geq$ 60 years admitted during 2020–2023 and having undergone lung cancer surgery, postoperative cardiopulmonary complications were examined using the CD classification as groups (CD grade I versus $\geq$ II), and the risk factors were analyzed using logistic regression and receiver operating characteristic (ROC) curves.

### Results

Of the 239 elderly patients, 29.3% had postoperative complications (CD $\geq$ II). Subgroup analysis revealed that patients aged $\geq$70 years had a higher rate of postoperative complications compared to those aged 60–69 years, however, this relationship was not statistically significant in the multivariable model (OR: 2.03, 95% CI: 0.95–4.36, p = 0.068). The CD grade $\geq$ II group had longer surgical time (p = 0.002), greater postoperative pulmonary complications (p < 0.001), and longer length of hospital stay (p < 0.001); CD grade $\geq$ II was more likely in patients with older age (odds ratio [OR]: 1.08, 95% confidence interval [CI]: 1.02–1.15, p = 0.011), COPD (OR: 4.41, 95% CI: 1.55–13.44, p = 0.005) and smoking history (OR: 2.85, 95% CI: 1.12–7.24, p = 0.028), having undergone pneumonectomy (OR: 14.89, 95% CI: 1.71–334.9, p = 0.045), and who converted to open thoracotomy (OR: 16.33, 95% CI: 2.13–169.71, p = 0.007). The area under the ROC curve was 0.81.

### Conclusions

Older age ($\geq$70 years) is associated with higher rates of postoperative complications (CD classification $\geq$ II) but is not an independent predictor when adjusting for other factors.

**Data Availability Statement:** All relevant data are within the manuscript and its Supporting Information files.

**Funding:** The author(s) received no specific funding for this work.

**Competing interests:** The authors have declared that no competing interests exist.

Comorbidities such as COPD and surgical factors, including pneumonectomy and conversion to thoracotomy, are significant contributors. These findings emphasize the need for comprehensive, multifactorial risk assessments to guide perioperative management and improve outcomes in elderly lung cancer patients.

## Introduction

Lung cancer is the most frequently occurring cancer and the leading cause of cancer-related deaths worldwide [1]. Pulmonary resection is key treatment for lung cancer but is often accompanied by significant postoperative complications, which may prolong hospital stay and incur greater medical expenses [2]. Surgical approaches for lung cancer, including video-assisted thoracoscopic surgery (VATS) and thoracotomy, offer distinct advantages and challenges. Previous studies [3, 4] suggest that VATS is superior to thoracotomy due to fewer postoperative complications.

Patients undergoing thoracic surgery were more likely to be older. Poor physiological changes are common in elderly patients with complex comorbidities, which could complicate major noncardiac surgery, especially thoracic surgery, and negatively impact outcomes [5]. The risk of cardiopulmonary complications increases with age [6] and the incidence, characteristics, and risk factors of postoperative cardiopulmonary complications in noncardiac surgery in elderly patients could be different from those in the general population. It is imperative to evaluate better risk indicators to improve preoperative cardiac risk prediction in elderly patients.

This study aims to fill this gap by identifying key predictors of severe postoperative complications (Clavien-Dindo classification grade $\geq$ II) in patients aged $\geq$60 years underwent lung cancer surgery. By analyzing patient-specific, surgical, and intraoperative factors, this study provides evidence to guide the selection of optimal surgical techniques, refine perioperative management, and improve outcomes for elderly patients. These findings contribute new insights into risk stratification and decision-making in this high-risk population.

## Materials and methods

### Institutional review board

This study was reviewed and approved by the Institutional Ethics Committee of the Faculty of Medicine, Prince of Songkla University, Songkhla, Thailand (REC.66-308-8-1) on September 15, 2023 and the requirement for patient consent was waived due to the nature of the study.

### Study design and setting

This retrospective cohort study was performed at the Songklanagarind Hospital, a tertiary care center in Southern Thailand. We screened the medical records of elderly patients $\geq$ 60 years who underwent elective lung cancer surgery between January 1, 2020, and December 31, 2023. The data for research purposes were accessed between on January 1, 2024 and January 31, 2024. The exclusion criterion was emergency lung surgery. We then retained postoperative cardiopulmonary complications and utilized the Clavien–Dindo classification [7] (Table 1) to grade the severity of postoperative complications.

**Data collection.** We examined the demographic characteristics of the patients, such as age, sex, body mass index (BMI), underlying diseases, preoperative serum albumin level, and

Table 1. Definition of the Clavien–Dindo classification [7].

| CD classifications | Definitions |
|---|---|
| Grade I | Any deviation from the normal postoperative course without the need for pharmacological treatment or surgical intervention. |
| Grade II | Complications requiring pharmacological treatment. |
| Grade IIIa | A complication requiring surgical or radiological intervention rather than general anesthesia. |
| Grade IIIb | A complication requiring surgical or radiological intervention under general anesthesia. |
| CD classifications | Definitions |
| Grade IVa | A life-threatening complication requiring intensive care management, including single-organ dysfunction. |
| Grade IVb | A life-threatening complication requiring intensive care management, including multiple organ dysfunction. |
| Grade V | Death |

preoperative pulmonary function tests (PFTs). We collected data on intraoperative and postoperative complications and categorized them as Clavien–Dindo grades I and $\geq$ II.

**Outcomes.** The primary outcomes of this study were the predictive factors associated with postoperative cardiopulmonary complications of Clavien–Dindo classification grade $\geq$ II.

## Statistical analysis

Continuous variables are presented as mean and standard deviation or median and interquartile range, as appropriate, with differences between groups determined using the t-test or rank-sum test, as appropriate. Categorical variables are expressed as numbers and percentages, and differences between groups were compared using the Chi-square or Fisher's exact test, as appropriate. Predictive factors were assessed using a logistic regression model and receiver operating characteristic (ROC) curve. Significant variables (p<0.20) were included in the multivariate logistic regression analysis with morbidity as the dependent variable. The R Studio software was used for statistical analysis. Statistical significance was set at $p < 0.05$.

## Results

### Clinical characteristics of the study cohort

The clinical characteristics of the 239 elderly patients who underwent elective lung cancer surgery are presented in Table 2. Of the total, 70 patients (29.3%) had postoperative complications of Clavien–Dindo classification $\geq$ II. The patients who had Clavien–Dindo grade $\geq$ II were 71 years old (interquartile range [IQR]: 66, 75.8) and 62.9% male; 25.7% had chronic obstructive pulmonary disease (COPD), with a median ratio of forced expiratory volume (FEV1) and forced vital capacity (FVC) of 74% (IQR: 65.8, 79.2), and mean predicted FEV1 of 62% (standard deviation [SD]: 18.2).

### Intraoperative and postoperative conditions

Patients who had postoperative complications (Clavien–Dindo classification $\geq$ II) underwent more frequent lobectomy (82.9% versus 75.7%, p = 0.023), pneumonectomy (5.7% versus 0.6%, p = 0.023). Conversion to open thoracotomy was also more common (10.2% versus 1.3%, p = 0.007) along with longer surgical time (130 min [SD: 42.6] versus 112.5 min, [SD:

**Table 2. Characteristics and the Clavien–Dindo classification patterns of the 239 elderly patients who underwent elective lung cancer surgery.**

| Patient characteristics | Clavien–Dindo grade ≥ II (N = 70, 29.3%) | Clavien–Dindo grade I (N = 169, 70.7%) | P-value |
|---|---|---|---|
| **Age (years),** median (IQR) | 71 (66, 75.8) | 69 (64, 73) | 0.034* |
| **Age interval** | | | 0.087 |
| **60–69 years** | 26 (37.1%) | 85 (50.3%) | |
| **≥ 70 years** | 44 (62.9%) | 84 (49.7%) | |
| **Male** | 44 (62.9%) | 76 (45%) | 0.018* |
| **Body weight,** median (IQR) | 59.5 (54, 70) | 60 (52, 69) | 0.774 |
| **Body height,** mean (SD) | 161.1 (7.8) | 158.7 (8.6) | 0.044* |
| **Body mass index (kg/m$^2$),** median (IQR) | 23.3 (21.1, 27) | 23.8 (21.9, 26.4) | 0.517 |
| **Ischemic heart disease** | 6 (8.6%) | 11 (6.5%) | 0.586 |
| **History of CVA** | 5 (7.1%) | 15 (8.9%) | 0.854 |
| **Hypertension** | 38 (54.3%) | 96 (56.8%) | 0.831 |
| **Dyslipidemia** | 33 (47.1%) | 86 (50.9%) | 0.7 |
| **Diabetes mellitus** | 13 (18.6%) | 29 (17.2%) | 0.941 |
| **Asthma** | 2 (2.9%) | 4 (2.4%) | 1.0 |
| **COPD** | 18 (25.7%) | 11 (6.5%) | <0.001* |
| **Restrictive lung disease** | 0 (0%) | 1 (0.6%) | 1.0 |
| **Renal insufficiency** | 16 (22.9%) | 35 (20.7%) | 0.845 |
| **Atrial fibrillation** | 6 (8.6%) | 5 (3%) | 0.086 |
| **Smoking** | 21 (30%) | 30 (17.8%) | 0.054 |
| **Metastasis** | 20 (28.6%) | 55 (32.5%) | 0.653 |
| **Pre-op hemoglobin (g/dL),** mean (SD) | 12.9 (1.7) | 12.5 (1.4) | 0.063 |
| **Pre-op albumin,** median (IQR) | 4.1 (3.8, 4.3) | 4.2 (4, 4.4) | 0.099 |
| **PNI score,** median (IQR) | 49.7 (47, 53.7) | 50.4 (47.7, 54.3) | 0.62 |
| **Pre-op GFR (mL/min/1.73 m$^2$),** median (IQR) | 78 (64.8, 92.8) | 80 (64, 92) | 0.934 |
| **Pulmonary function tests** | | | |
| **FVC,** median (IQR) | 2.4 (1.8, 2.9) | 2.4 (2, 2.8) | 0.96 |
| **FEV1,** mean (SD) | 1.7 (0.5) | 1.9 (0.5) | 0.086 |
| **%FEV1,** mean (SD) | 80.8 (20.7) | 91.4 (17) | <0.001* |
| **FEF 25–75%,** mean (SD) | 1.4 (0.7) | 1.7 (0.6) | 0.001* |
| **FEV1/FVC (%),** median (IQR) | 74 (65.8, 79.2) | 78 (73, 83) | 0.001* |
| **% Predicted FEV1,** mean (SD) | 62 (18.2) | 70.8 (14) | <0.001* |
| **ASA class,** median (IQR) | 2 (2, 3) | 2 (2, 3) | 0.671 |

All data are presented as numbers (%) unless indicated otherwise.

*P-value<0.05 indicates statistical significance.

IQR, interquartile range; CVA, cerebrovascular accident; COPD, chronic obstructive pulmonary disease; SD, standard deviation; PNI, prognostic nutritional index; GFR, glomerular filtration rate; FVC, forced vital capacity; FEV1, forced expiratory volume; FEF, forced expiratory flow; ASA, American Society of Anesthesiologists.

37.2], p = 0.002). These patients exhibited higher rate of postoperative pulmonary complications (p < 0.001), and longer hospital stay (10 days [IQR: 7.2, 14.8] versus 7 days [IQR 6, 8], p < 0.001) (Table 3). Open thoracotomy was associated with increased incidence of postoperative pulmonary complications (91.2% versus 77.6%, p = 0.012) and longer hospital stays (8 days [IQR: 6, 11] versus 6 days [IQR: 5, 7], p < 0.001) compared to VATS procedures (Table 4).

**Table 3. Intraoperative and postoperative conditions and the Clavien–Dindo classifications in the 239 elderly patients.**

| Intraoperative and postoperative conditions | Clavien–Dindo grade ≥ II (N = 70, 29.3%) | Clavien–Dindo grade I (N = 169, 70.7%) | P-value |
|---|---|---|---|
| **GA with epidural block** | 48 (68.6%) | 103 (60.9%) | 0.335 |
| **GA with paravertebral block** | 3 (4.3%) | 10 (5.9%) | 0.761 |
| **GA with erector spinae block** | 5 (7.1%) | 13 (7.7%) | 1.000 |
| **GA with intercostal block** | 8 (11.4%) | 20 (11.8%) | 1.000 |
| **Operations** | | | 0.137 |
| VATS | 12 (17.1%) | 46 (27.2%) | |
| Open thoracotomy | 58 (82.9%) | 123 (72.8%) | |
| **Extent of surgery** | | | 0.023* |
| Segmentectomy | 0 (0%) | 5 (3%) | |
| Lobectomy | 58 (82.9%) | 128 (75.7%) | |
| Pneumonectomy | 4 (5.7%) | 1 (0.6%) | |
| Metastasectomy | 8 (11.4%) | 30 (17.8%) | |
| Wedge resection | 0 (0%) | 5 (3%) | |
| **Convert to open thoracotomy** | 6 (10.2%) | 2 (1.3%) | 0.007* |
| **Surgical time (min),** mean (SD) | 130 (42.6) | 112.5 (37.2) | 0.002* |
| **Blood loss (ml),** median (IQR) | 50 (50, 137.5) | 50 (20, 100) | 0.003* |
| **Post-op pulmonary complications** | 69 (28.9%) | 141 (59%) | |
| **Post-op hypoxia** | 17 (24.6%) | 4 (2.8%) | <0.001* |
| **Atelectasis** | 22 (31.9%) | 16 (11.3%) | <0.001* |
| **Pneumonia** | 11 (15.9%) | 1 (0.7%) | <0.001* |
| **Re-intubation** | 8 (11.6%) | 0 (0%) | <0.001* |
| **Remained intubation > 24 hours** | 10 (14.5%) | 0 (0%) | <0.0018 |
| **Pleural effusion** | 15 (21.7%) | 12 (8.5%) | 0.013* |
| **Pneumothorax** | 18 (26.1%) | 21 (14.9%) | 0.077 |
| **Post-op cardiac complications** | 30 (12.6%) | 5 (2.1%) | |
| **Arrhythmia** | 13 (43.3%) | 1 (20%) | 0.627 |
| **Hypotension** | 19 (63.3%) | 0 (0%) | 0.013* |
| **On inotrope** | 11 (36.7%) | 0 (0%) | 0.157 |
| **Congestive heart failure** | 4 (13.3%) | 0 (0%) | 1.000 |
| **Myocardial infarction** | 1 (3.3%) | 0 (0%) | 1.000 |
| **Blood transfusion requirement** | 20 (66.7%) | 3 (60%) | 1.000 |
| **Reoperation** | 6 (8.6%) | 0 (0%) | <0.001* |
| **Length of hospital stay (days),** median (IQR) | 10 (7.2, 14.8) | 7 (6, 8) | <0.001* |
| **ICU admission** | 15 (21.4%) | 0 (0%) | <0.001* |
| **In-hospital mortality** | 4 (5.7%) | 0 (0%) | 0.007* |

All data are presented as numbers (%) unless indicated otherwise.

*P-value<0.05 indicates statistical significance.

GA, general anesthesia; VATS, video-assisted thoracic surgery; SD, standard deviation; IQR, interquartile range; ICU, intensive care unit; op, operation.

## Factors associated with postoperative complications (Clavien–Dindo classification ≥ II)

The factors associated with postoperative complications (Clavien–Dindo classification ≥ II) included older age (odds ratio [OR]: 1.08, 95% confidence interval [CI]: 1.02–1.15, p = 0.011), presence of COPD (OR: 4.41, 95% CI: 1.55–13.44, p = 0.005), smoking history (OR: 2.85, 95%

**Table 4. Intraoperative and postoperative conditions and types of surgery in the 239 elderly patients.**

| Intraoperative and postoperative conditions | VATS (N = 58, 24.3%) | Open thoracotomy (N = 181, 75.7%) | P-value |
|---|---|---|---|
| **Intraoperative conditions** | | | |
| **Convert to open thoracotomy** | 8 (17%) | 0 (0%) | <0.001* |
| **Massive blood loss** | 1 (2.1%) | 0 (0%) | 0.226 |
| **Hypotension** | 30 (63.8%) | 137 (85.1%) | 0.003* |
| **Bradycardia** | 27 (57.4%) | 58 (36%) | 0.014* |
| **Desaturation** | 10 (21.3%) | 6 (3.7%) | <0.001* |
| **Remained intubation** | 2 (4.3%) | 5 (3.1%) | 0.657 |
| **Postoperative conditions** | | | |
| **Clavien–Dindo classifications,** median (IQR) | 1 (1, 1) | 1 (1, 2) | 0.06 |
| **Pulmonary complications** | 45 (77.6%) | 165 (91.2%) | 0.012* |
| **Hypoxia** | 1 (2.2%) | 20 (12.1%) | 0.052 |
| **Atelectasis** | 8 (17.8%) | 30 (18.2%) | 1.000 |
| **Pneumonia** | 2 (4.4%) | 10 (6.1%) | 1.000 |
| **Pneumothorax** | 12 (26.7%) | 27 (16.4%) | 0.174 |
| **Reintubation** | 1 (2.2%) | 7 (4.2%) | 1.000 |
| **Pleural effusion** | 1 (2.2%) | 26 (15.8%) | 0.031* |
| **Cardiac complications** | 5 (8.6%) | 30 (16.6%) | 0.201 |
| **Arrhythmia** | 1 (20%) | 13 (43.3%) | 0.627 |
| **Hypotension** | 4 (80%) | 15 (50%) | 0.347 |
| **Myocardial infarction** | 0 (0%) | 1 (3.3%) | 1.000 |
| **On inotrope** | 3 (60%) | 8 (26.7%) | 0.297 |
| **Congestive heart failure** | 0 (0%) | 4 (13.3%) | 1.000 |
| **Unplanned ICU** | 0 (0%) | 8 (26.7%) | 0.315 |
| **Length of hospital stay (days),** median (IQR) | 6 (5, 7) | 8 (6, 11) | <0.001* |
| **Admitted ICU** | 1 (1.7%) | 14 (7.7%) | 0.126 |
| **In-hospital mortality** | 0 (0%) | 4 (2.2%) | 0.575 |

All data are presented as numbers (%) unless indicated otherwise.

*P-value<0.05 indicates statistical significance.

VATS, video-assisted thoracic surgery; IQR, interquartile range; ICU, intensive care unit.

CI: 1.12–7.24, p = 0.028), undergoing pneumonectomy (OR: 14.89, 95% CI: 1.71–334.9, p = 0.045), and conversion to open thoracotomy (OR: 16.33, 95% CI: 2.13–169.71, p = 0.007) (Table 5). Subgroup analysis compared patients aged 60–69 years and those aged ≥ 70 years. The analysis revealed that patients aged ≥ 70 years were more likely to have postoperative complications in the univariate model (OR 1.92, 95% CI: 1.02–3.63, p = 0.045). However, after adjusting for other risk factors in the multivariate model, this association was not statistically significant (OR 2.03, 95% CI: 0.95–4.36, p = 0.068) (Table 5). The area under the ROC curve of the ability of risk factors to predict the postoperative complications (Clavien–Dindo classification ≥ II) was 0.81.

## Discussion

Our study investigated the outcomes of lung cancer surgery in elderly patients, focusing on postoperative complications and mortality rates. We observed an in-hospital mortality rate of 1.7% after lung cancer surgery. Benker et al. [8] reported an overall operative mortality rate of

**Table 5. Factors associated in the univariate and multivariate regression models with the Clavien–Dindo classification $\geq$ II.**

| Factors | Univariate | | Multivariable | |
|---|---|---|---|---|
| | OR (95% CI) | P-value | OR (95% CI) | P-value |
| Age (years) | 1.06 (1.01, 1.11) | 0.019 | **1.08 (1.02, 1.15)** | **0.011**[*] |
| Age $\geq$ 70 VS 60–69 years | 1.92 (1.02, 3.63) | 0.045 | 2.03 (0.95, 4.36) | 0.068 |
| Male | 2.28 (1.22, 4.32) | 0.010 | 2 (0.61, 6.68) | 0.251 |
| Body height | 1.03 (0.99, 1.07) | 0.105 | | |
| COPD | 6.29 (2.56, 16.61) | <0.001 | **4.41 (1.55, 13.44)** | **0.005**[*] |
| Atrial fibrillation | 0.36 (0.1, 1.35) | 0.119 | | |
| Smoking | 2.53 (1.23, 5.16) | 0.011 | **2.85 (1.12, 7.24)** | **0.028**[*] |
| Pre-op FEV1 | 0.55 (0.24, 1.25) | 0.153 | | |
| Pre-op FEV1/FVC | 0.95 (0.92, 0.99) | 0.011 | 0.998 (0.923, 1.079) | 0.959 |
| Predicted FEV1 | 0.97 (0.94, 0.99) | 0.012 | 0.97 (0.91, 1.04) | 0.424 |
| Open thoracotomy (ref = VATs) | 1.33 (0.64, 2.95) | 0.458 | | |
| Extent of surgery (ref = lobectomy) | | | | |
| Segmentectomy | 0 (0, inf) | 0.991 | | |
| Pneumonectomy | 9.96 (1.43, 197.54) | 0.042 | | |
| Metastasectomy | 0.48 (0.17, 1.16) | 0.127 | **14.89 (1.71, 334.9)** | **0.045**[*] |
| Wedge resection | 0 (0, inf) | 0.993 | | |
| Surgical time | 1.01 (1.01, 1.02) | <0.001 | 1.01 (1, 1.02) | 0.059 |
| Blood loss | 1.0017 (1.0005, 1.0035) | 0.021 | 1.0004 (0.9987, 1.0025) | 0.707 |
| Conversion to open thoracotomy | 8.65 (1.92, 60.31) | 0.010 | **16.33 (2.13, 169.71)** | **0.007**[*] |
| Intra-op hypotension | 2.11 (0.92, 5.48) | 0.095 | | |
| Intra-op vasopressor | 0.29 (0.07, 1.12) | 0.07 | | |

[*]P-value<0.05 indicates statistical significance.

OR, odds ratio; CI, confidence interval; COPD, chronic obstructive pulmonary disease; Pre-op, pre-operative; WBC, white blood cell; FEV1, forced expiratory volume; FVC, forced vital capacity; ref, reference; Intra-op, intraoperative.

1.56%, similar to our finding. In contrast, Detillon et al. [9] found a higher operative mortality rate by 2.1% in patients $\geq$ 60 years with lung cancer resections compared to our data.

## Postoperative complications

We found that 28.9% of patients experienced postoperative pulmonary complications and 12.6% experienced cardiac complications classified as Clavien–Dindo grade $\geq$ II. These rates are higher compared to the previous studies reporting 10–15.4% for pulmonary complications in elderly patients undergoing lung resection [10, 11]. Similarly, Calado et al. [12] reported lower cardiac complications rates (8.2%) compared to our findings. In contrast, Srisomboon et al. [13] reported higher rate of respiratory (25%) and lower rates of cardiac complications (6%) in elderly patients undergoing thoracoscopic surgery.

## Factors associated with postoperative complications

The overall postoperative pulmonary and cardiac complications of Clavien–Dindo grade $\geq$ II were associated with age, male sex, body height, COPD, lower preoperative PFTs (% pre-operative $FEV_1$, $FEF_{25-75\%}$, $FEV_1/FVC$, % predicted $FEV_1$), lobectomy, pneumonectomy, surgical time, intraoperative blood loss, and conversion from VATS to open thoracotomy. Age,

coexistence of COPD, smoking, pneumonectomy, and conversion from VATS to open thoracotomy were significant predictors in the multivariable logistic model.

We found that patients aged ≥70 years were more likely to experience postoperative complications (Clavien-Dindo classification ≥2) in the univariate analysis. However, this association was attenuated in the multivariable model, suggesting that age alone may not be an independent predictor of postoperative complications. Similarly, previous studies have reported that patients aged ≥ 75 years did not exhibit increased surgical morbidity or mortality when appropriately selection for lung resection [14, 15]. This suggests that age, while an important factor, may act as a surrogate for other predictors such as comorbidities (e.g., COPD) or intraoperative challenges. These findings highlight the importance of comprehensive risk assessment that integrates patient- and surgery-specific factors to optimize outcomes in elderly lung cancer patients. In contrast, other studies have found that age is a predisposing factor for the development of postoperative complications after lung resection [8, 16]. These differing findings underscore the need for individualized patient evaluation, balancing the physiological effects of aging with other clinical factors to refine surgical decision-making.

Scarci et al. [17] found that COPD is a predisposing factor to postoperative complications, similar to our study. However, Benker et al. [8] found that COPD was not associated with postoperative pulmonary complications. A previous study [18] reported that the coexistence of asthma was a significant risk factor for postoperative complications, which was not observed in our study.

Ogawa et al. [15] found that postoperative complications in elderly lung cancer patients are dependent on factors such as smoking and blood loss during surgery, which is similar to our finding.

We found that postoperative complications were more likely in men than in women, consistent with previous study findings [14, 17–19].

Previous studies [8, 10] have found that American Society of Anesthesiologists (ASA) classification ≥ 3 and lower BMI were independently associated with postoperative pulmonary complication risk, while our study found that patients with a greater body height had increased postoperative complications and that the ASA classification was not related to postoperative complications.

Similar to our study, many studies [8, 17, 18] have found that low preoperative PFTs, especially FEV1, are significant risk factors for postoperative complications. Choi et al. [20] found that patients with $ppoFEV_1$ < 40% showed higher rates of pulmonary complications (13% versus 24%, p = 0.014) compared to patients with ppoFEV1 > 40%. Our study found that ppoFEV1 < 62 ± 18.2% showed higher rates of postoperative complications. However, we found that PFTs themselves were not effective predictors of postoperative complications of Clavien–Dindo classification grade ≥ II after lung cancer surgery, which may be due to insufficient data.

In this study, we found that the type of operation was not associated with the overall postoperative complications of Clavien–Dindo classification grade ≥ II (p = 0.06). However, open thoracotomy significantly increased postoperative pulmonary complications compared to VATS but did not significantly increase postoperative cardiac complications. Similarly, previous studies found that VATS in patients > 60 years with lung cancer reduced morbidity compared with thoracotomy [4, 9, 21, 22]. We found that VATS reduced the length of hospital stay compared to thoracotomy, which is consistent with results from previous studies [4, 21, 22]. Performing VATS, when possible, could reduce the incidence of postoperative complications, especially pulmonary complications, in elderly patients. However, we found that conversion from VATS to open thoracotomy significantly increased the incidence of postoperative complications. This may be caused by surgical complications or bleeding that leads to increased

surgical time, blood loss, and transfusion of blood products, which then leads to increased postoperative complications.

This study found that performing lobectomy and pneumonectomy significantly increased postoperative complications compared with segmentectomy, wedge resection, and metastectomy. We found that pneumonectomy, as an independent predictive factor, increased postoperative complications when compared with lobectomy. A previous study demonstrated that operative procedures, such as lobectomy or a greater resection, predicted morbidity and mortality in patients > 70 years [10, 18, 23]. In contrast, Berry et al. [4] found that short-term outcomes were equivalent for wedge resections and more extensive resections. These findings suggest that overall postoperative complications in elderly patients increase after surgeries with a prolonged surgical time. Other studies found that a longer surgery time was a predictor of postoperative complications in elderly patients after lung cancer surgery [11, 24, 25]. Okada et al. [11] found that prolonged surgical time increased pulmonary complications in elderly patients; however, this was not found in our study. Hence, clinicians should consider adopting surgical approaches that can be performed without a prolonged surgical time.

Increasing age is associated with higher rates of comorbidities and increased malnutrition-related mortality [24]. Malnutrition is a risk factor for morbidity and mortality in surgical patients, and more surgical comorbidities are seen in elderly patients than in younger patients [25]. Poor nutrition, characterized by low albumin levels, was also an independent predictor of postoperative complications in octogenarian patients in a Japanese nationwide study [26]. Okada et al. [11] found that lower prognostic nutritional index (PNI) values were significantly associated with postoperative pulmonary complications in elderly patients with non-small cell lung cancer. In contrast, our study found that serum albumin levels and PNI values were not associated with postoperative complications in elderly patients. This may be due to incomplete data because some patients did not have serum albumin levels and PNI values could not be calculated.

## Limitations

This study had some limitations. First, this was a retrospective cohort study; hence, missing database records may have compromised the analysis. Some information was unavailable in the registry of the facility, such as serum albumin levels, which were not routine tests. Information about the main causes of cardiopulmonary complications, which were not always available including the PFTs, were not routinely examined during the coronavirus disease 2019 pandemic or in patients who underwent metastatectomy or wedge resection. Second, postoperative complications were extracted from data records, which may have missed some data owing to the retrospective nature of the study. Third, this single-institution study was not adequate for making universal conclusions. Finally, no long-term complications were observed.

## Conclusion

In this study, postoperative complications (Clavien-Dindo grade $\geq$ II) were observed in nearly one-third of elderly patients undergoing lung cancer surgery. While older age ($\geq$70 years) was associated with a higher rate of postoperative complications in the univariate analysis, this relationship was not significant in the multivariable model, suggesting that age alone is not an independent predictor of adverse outcomes. Instead, significant contributors to complications included comorbidities such as COPD, smoking history, and surgical factors like pneumonectomy and conversion to open thoracotomy.

These findings highlight the importance of a comprehensive risk assessment that integrates patient- and surgery-specific factors. Such an approach can guide perioperative planning and

decision-making, enabling optimization of outcomes in this high-risk population. Further research is needed to refine risk prediction models and explore strategies to mitigate these risks, particularly in older patients with significant comorbidities or undergoing complex surgical procedures.

## Supporting information

**S1 Table. Characteristics patterns of 239 elderly patients who underwent lung cancer surgery.**
(CSV)

**S1 Data. Factors associated with Clavien-Dindo classification $\geq$ II.**
(CSV)

## Acknowledgments

The authors thank Editage for their assistance in editing the language of this research project and for ensuring the clarity and coherence of our manuscript.

## Author Contributions

**Conceptualization:** Mantana Saetang, Thitikan Kunapaisal.

**Data curation:** Mantana Saetang, Thitikan Kunapaisal, Parin Boonthum, Bussarin Sriyanaluk, Kanjana Nuanjun.

**Formal analysis:** Mantana Saetang, Thitikan Kunapaisal.

**Investigation:** Mantana Saetang, Thitikan Kunapaisal.

**Methodology:** Mantana Saetang, Thitikan Kunapaisal.

**Project administration:** Mantana Saetang, Thitikan Kunapaisal, Wirat Wasinwong.

**Resources:** Thitikan Kunapaisal, Parin Boonthum.

**Software:** Mantana Saetang, Thitikan Kunapaisal.

**Supervision:** Mantana Saetang, Wirat Wasinwong.

**Validation:** Mantana Saetang, Thitikan Kunapaisal.

**Visualization:** Mantana Saetang.

**Writing – original draft:** Mantana Saetang, Thitikan Kunapaisal.

**Writing – review & editing:** Mantana Saetang, Thitikan Kunapaisal, Wirat Wasinwong, Parin Boonthum, Bussarin Sriyanaluk, Kanjana Nuanjun.

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
