## [Decision Letter · Decision Letter 0]

6 Nov 2024

PONE-D-24-44073Predictors associated with Clavien–Dindo complications in lung cancer surgery: a retrospective cohort study.PLOS ONE

Dear Dr. Kunapaisal,

Thank you for submitting your manuscript to PLOS ONE. After careful consideration, we feel that it has merit but does not fully meet PLOS ONE’s publication criteria as it currently stands. Therefore, we invite you to submit a revised version of the manuscript that addresses the points raised during the review process.

We look forward to receiving your revised manuscript.

Kind regards,

Luca Bertolaccini, M.D., Ph.D.

Academic Editor

PLOS ONE

2. We note that your Data Availability Statement is currently as follows: [All relevant data are within the manuscript and its supporting information files.]

Additional Editor Comments:

The reviewers have emphasised issues that require a careful and thorough manuscript revision.

No commitment to publication can be made at this point.

Reviewers' comments:

Reviewer's Responses to Questions

**Comments to the Author**

1. Is the manuscript technically sound, and do the data support the conclusions?

Reviewer #1: Partly

Reviewer #2: Partly

2. Has the statistical analysis been performed appropriately and rigorously? 

Reviewer #1: I Don't Know

Reviewer #2: Yes

3. Have the authors made all data underlying the findings in their manuscript fully available?

Reviewer #1: No

Reviewer #2: Yes

4. Is the manuscript presented in an intelligible fashion and written in standard English?

Reviewer #1: Yes

Reviewer #2: No

5. Review Comments to the Author

Reviewer #1: Dr. Saetang and colleagues reported an interesting paper regarding postoperative complications and possible associated risk factors.

The article is of possible interest, but I have some comments.

- First of all: most of the findings of the paper are somewhat already known. Authors should focus on what is brought as clinical new message by their own paper.

- Some words should be revised (e.g. in table: "mode of surgery" should be changed with "extent of surgery"; "Operations" with "surgical technique")

- In the Material and methods, definition of Clavien-Dindo complications could be put in a table rather then written in the text

- Results of complications according to the surgical technique (minimally invasive/open) or extent of surgical resection (segmentectomy/lobectomy/pneumonectomy/extended resection) should be reported in the text with more details and not only in the tables.

- Moreover I would try to perform a sub analysis of "best patients" (e.g. VATS, Lobectomy or sublobar resections, no conversion) and try to see if some of the other risk factors remain significant in this subgroup.

- Another important factors are introperative complications, like severe bleeding. This should be taken into account in data analysis.

- the ROC analysis I think it is unnecessary.

Reviewer #2: Good statistical analysis but I suggest to define elderly patient and divide younger from older patients in order to obtain more correct results. Obiouvsly elderly patients have much more postoperative complications.

6. PLOS authors have the option to publish the peer review history of their article (what does this mean?). If published, this will include your full peer review and any attached files.

Reviewer #1: **Yes: **Pietro Bertoglio

Reviewer #2: No

---

## [Author Response · Author response to Decision Letter 0]

19 Nov 2024

I uploaded minimal data set in supporting information files.

---

## [Decision Letter · Decision Letter 1]

9 Dec 2024

Predictors associated with Clavien–Dindo complications in lung cancer surgery: a retrospective cohort study.

PONE-D-24-44073R1

Dear Dr. Kunapaisal,

We’re pleased to inform you that your manuscript has been judged scientifically suitable for publication and will be formally accepted for publication once it meets all outstanding technical requirements.

Kind regards,

Luca Bertolaccini, M.D., Ph.D.

Academic Editor

PLOS ONE

Additional Editor Comments (optional):

Reviewers' comments:

Reviewer's Responses to Questions

**Comments to the Author**

1. If the authors have adequately addressed your comments raised in a previous round of review and you feel that this manuscript is now acceptable for publication, you may indicate that here to bypass the “Comments to the Author” section, enter your conflict of interest statement in the “Confidential to Editor” section, and submit your "Accept" recommendation.

Reviewer #1: All comments have been addressed

2. Is the manuscript technically sound, and do the data support the conclusions?

Reviewer #1: Partly

3. Has the statistical analysis been performed appropriately and rigorously? 

Reviewer #1: I Don't Know

4. Have the authors made all data underlying the findings in their manuscript fully available?

Reviewer #1: Yes

5. Is the manuscript presented in an intelligible fashion and written in standard English?

Reviewer #1: Yes

6. Review Comments to the Author

Reviewer #1: Authors addressed all the comments that I raised and I think the quality of the paper has been improved. I have no further comments.

7. PLOS authors have the option to publish the peer review history of their article (what does this mean?). If published, this will include your full peer review and any attached files.

Reviewer #1: **Yes: **Pietro Bertoglio

---

## [Editor Report · Acceptance letter]

12 Dec 2024

PONE-D-24-44073R1 

PLOS ONE

Dear Dr. Kunapaisal, 

I'm pleased to inform you that your manuscript has been deemed suitable for publication in PLOS ONE. Congratulations! Your manuscript is now being handed over to our production team.

Kind regards, 

on behalf of

Dr. Luca Bertolaccini 

Academic Editor

PLOS ONE